# A Fault Diagnosis Method of Bogie Axle Box Bearing Based on Spectrum Whitening Demodulation

**DOI:** 10.3390/s20247155

**Published:** 2020-12-14

**Authors:** Zejun Zheng, Dongli Song, Xiao Xu, Lei Lei

**Affiliations:** Key Laboratory of Traction Power, Southwest Jiaotong University, Chengdu 610031, China; zhengjuner@my.swjtu.edu.cn (Z.Z.); XuXiao@my.swjtu.edu.cn (X.X.); leilei309@my.swjtu.edu.cn (L.L.)

**Keywords:** fault diagnosis, axle box bearing, frequency domain whitening, energy operator, feature extraction

## Abstract

The axle box bearing of bogie is one of the key components of the rail transit train, which can ensure the rotary motion of wheelsets and make the wheelsets adapt to the conditions of uneven railways. At the same time, the axle box bearing also exposes most of the load of the car body. Long-time high-speed rotation and heavy load make the axle box bearing prone to failure. If the bearing failure occurs, it will greatly affect the safety of the train. Therefore, it is extremely important to monitor the health status of the axle box bearing. At present, the health status of the axle box bearing is mainly monitored by vibration information and temperature information. Compared with the temperature data, the vibration data can more easily detect the early fault of the bearing, and early warning of the bearing state can avoid the occurrence of serious fault in time. Therefore, this paper is based on the vibration data of the axle box bearing to carry out adaptive fault diagnosis of bearing. First, the AR model predictive filter is used to denoise the vibration signal of the bearing, and then the signal is whitened in the frequency domain. Finally, the characteristic value of vibration data is extracted by energy operator demodulation, and the fault type is determined by comparing with the theoretical value. Through the analysis of the constructed simulation signal data, the characteristic parameters of the data can be effectively extracted. The experimental data collected from the bearing testbed of high-speed train are analyzed and verified, which further proves the effectiveness of the feature extraction method proposed in this paper. Compared with other axle box bearing fault diagnosis methods, the innovation of the proposed method is that the signal is denoised twice by using AR filter and spectrum whitening, and the adaptive extraction of fault features is realized by using energy operator. At the same time, the steps of setting parameters in the process of feature extraction are avoided in other feature extraction methods, which improves the diagnostic efficiency and is conducive to use in online monitoring system.

## 1. Introduction

Fault diagnosis is a kind of technology to understand and master the state of the machine in the process of operation, determine whether the state is normal or abnormal, find the fault and its causes, and predict the development trend of the fault. It is one of the most important measures to maintain the safe and stable operation of the equipment. The main tasks of fault diagnosis are fault detection, fault type judgment, fault location, and fault recovery.

By the end of 2019, the operating mileage of China’s high-speed railway has exceeded 35,000 km, more than two-thirds of the total mileage of the world’s high-speed railway. At the same time, the operation speed has been continuously improved, and the actual operating speed of some railway lines has reached 350 km/h. The increase of speed and mileage means that the axle box bearing will be subject to more severe and complex impact in operation. According to the statistics of relevant data, the number of bogie axle box bearing failures of rail transit vehicles is about half of the total number of bogie bearing (including axle box bearing, motor bearing and gearbox bearing) failures, and the main fault occurs in the outer ring of bearing. The statistics of bogie bearing failure are shown in Figure 1 [1].

At present, most scholars study the health status of axle box bearing from the vibration data and temperature data as the data basis, and some scholars start with the sound signal collected by the trackside detection equipment, such as the trackside acoustic device system (TADS).

Using the research method of axle box bearing fault diagnosis and early warning based on temperature data, the temperature time series data obtained by bearing temperature tracking for a long time are analyzed. Liu et al. [2] constructed the multilayer long short-term memory-isolation forest (MLSTM-iForest) network, which can predict the temperature data of the later period through the existing temperature time series data, so as to realize the early warning of bearing fault. Cheng et al. [3] proposed an abnormal index model condition monitoring method based on local outlier factor (LOF) algorithm, which can detect the potential fault of axle box bearing in advance.

Bearing fault diagnosis and analysis methods by using vibration data are mainly divided into time-domain analysis, frequency-domain analysis, and time-frequency-domain analysis. The time-domain analysis method is mainly used to solve the time domain amplitude parameters of vibration signal data, such as root mean square value (RMS), peak value, peak-to-average power ratio (PAPR), kurtosis, etc. The results show that the bearing surface damage fault is sensitive to the peak value index and the wear type fault is sensitive to the root mean square value. Cai et al. [4] processed the signal through ensemble empirical mode decomposition (EEMD), introduced the coefficient of variation, and constructed two steady-state indexes from the preprocessed vibration data in time-domain and frequency-domain, respectively, to distinguish different health states of bearings. Cheng et al. [5] overcame the shortcomings of the original multipoint optimal minimum entropy deconvolution adjusted (MOMEDA) method through the improved MOMEDA method, and effectively improved the accuracy of fault diagnosis of axle box bearing. Chen et al. [6] proposed that periodicity detection techniques (PDTs) assist maximum correlated kurtosis deconvolution (MCKD), MOMEDA, and cyclostationarity blind deconvolution (CYCBD) to enhance bearing fault features, which can improve the accuracy of fault identification. Yan et al. [7] proposed a novel early fault detection strategy based on an enhanced scale morphological-hat product filtering which can effectively extract periodic pulses related to bearing fault and realize early fault detection of bearing. Li et al. [8] used morphological component analysis (MCA) method and improved its sparse representation dictionary to effectively separate the fault characteristics of high-speed train axle box bearing under impact excitation. Wang et al. [9] proposed a bearing fault diagnosis method based on based on frequency-domain energy feature reconstruction (EFR) and composite multiscale permutation entropy (CMPE), and the bearing characteristic parameters are extracted and classified, which provides a new solution for condition monitoring and fault diagnosis of axle-box bearings.

The time-frequency analysis methods mainly include short-time Fourier transform and wavelet decomposition, which can identify the fault and determine the time period of the fault. Wu et al. [10] used wavelet analysis and FastICA-independent component analysis based on negative entropy to denoise the vibration signal of rolling bearing, and verified the effectiveness of the method. Song et al. [11] proposed a fault diagnosis scheme for high pressure common rail injector based on improved fruit fly optimization algorithm-variational mode decomposition (IFOA-VMD) and hierarchical dispersion entropy (HDE), which solved the non-adaptive problem of VMD and could also be used in the bearing fault diagnosis. Li et al. [12] obtained time-frequency spectrum samples through short-time Fourier transform of bearing vibration signal, and improved fault diagnosis accuracy by training neural network.

In order to effectively extract the characteristic parameters of axle box bearing vibration data, realize adaptive fault diagnosis and improve the diagnosis efficiency of on-line monitoring, a method combining AR prediction filter with spectrum whitening is proposed, and the characteristic parameters in frequency domain are extracted by energy operator demodulation. Through this method, the fault feature information of axle box bearing is obtained, and the fault location is determined by comparing with the theoretical fault frequency of bearing elements. After analyzing the simulation data, the characteristic frequency of the data can be extracted effectively. Finally, the test data are used to verify the effectiveness of the fault diagnosis method proposed in this paper and has a good diagnosis effect.

## 2. A Brief Description of the Methods for Vibration Data

### 2.1. AR Model Filter

The autoregressive (AR) model is a statistical method to deal with time series. It uses the values of the same variable in different periods to predict the present values and assumes that they are linear [13]. Therefore, it is called autoregressive. The mathematical expression of AR model is shown in Equation (1).
(1)$$X_{t}=c+\sum_{i=1}^{p}\varphi_{i}X_{t-i}+\varepsilon_{t}$$

In Equation (1), c is a absolute term, $\varphi_{i}$ is an autocorrelation coefficient, and $\varepsilon_{t}$ is the random error value (its average is 0, the standard deviation is the fixed value), which is white noise. The meaning of this equation can be expressed as that the expected value of *X* is composed of a linear combination of one or several previous time values, plus absolute term and random error.

In time series analysis, the AR model is often used for parameter estimation, and $\varepsilon_{t}$ keeps approaching the perfect white noise signal model by continuously selecting appropriate parameters. There are several methods to estimate the parameters of the AR model, such as the Maximum Entropy Method, Covariance Method, Autocorrelation Method, Burg Method, and Yule–Walker Method. The AR model parameter estimation method is used to obtain the best parameters of the model and realize the linear filtering of the original time domain signal. The Yule–Walker method is a classical AR model parameter estimation algorithm. The Yule–Walker equation is an equation describing the relationship between the parameters of autoregressive sequence and its covariance function. The common equation set form is shown in Equation (2) [13,14]:(2)$$\left[\begin{array}{c} \gamma_{1}\\ \gamma_{2}\\ \gamma_{3}\\ \vdots \\ \gamma_{p} \end{array}\right]=\left[\begin{array}{ccccc} \gamma_{0} & \gamma_{-1} & \gamma_{-2} & \cdots & \gamma_{1-p}\\ \gamma_{1} & \gamma_{0} & \gamma_{-1} & \cdots & \gamma_{2-p}\\ \gamma_{2} & \gamma_{1} & \gamma_{0} & \cdots & \gamma_{3-p}\\ \vdots & \vdots & \vdots & \ddots & \vdots \\ \gamma_{p-1} & \gamma_{p-2} & \gamma_{p-3} & \cdots & \gamma_{0} \end{array}\right]\left[\begin{array}{c} \varphi_{1}\\ \varphi_{2}\\ \varphi_{3}\\ \vdots \\ \varphi_{4} \end{array}\right]$$

In Equation (2), ${\left[\begin{array}{cccc} \gamma_{1} & \gamma_{2} & \cdots & \gamma_{p} \end{array}\right]}^{T}$ is the parameter estimate of the AR model.

The linear filter of AR model can be established by solving the parameter estimates of AR model. The filtering process of AR model filter used in this paper is as follows.
(1).Establish the AR model of different orders of original time-domain signals.(2).Estimate the parameter values of the AR model of different order by Yule–Walker equation.(3).Create linear filters corresponding to different orders and obtain filtered time-domain parameters.(4).Solve the kurtosis values of filtered signals by different filters and select the signal with the maximum kurtosis value as the best filtered signal.

The AR filtering process is shown in Figure 2.

### 2.2. Whitening of Time Domain Signal Spectrum

Frequency domain whitening of time domain signal means frequency domain editing of signal. The common editing method is Cepstrum Editing Pre-Whitening (CPW) [15,16,17,18]. This method whitens the cepstrum of original time domain signal and sets all the values except the first cepstrum line to zero in real Cepstrum to realize the whitening of each frequency component in signal frequency domain. Whitening can effectively remove the relatively discrete frequency components in the original signal, reduce the noise component of the signal, highlight the main characteristic frequency, and demodulate the resonance effect in the original signal. The flow chart of the CPW method is shown in Figure 3.

In the CPW algorithm, the original time domain signal needs Fourier transform to get the frequency domain amplitude spectrum and phase spectrum. Through resolving and editing the cepstrum of the amplitude spectrum, the cepstrum frequency domain amplitude spectrum is obtained, which is combined with the original phase spectrum. Through Fourier inverse transformation, the whitened time domain signal can be obtained.

Assume that the original time-domain discrete signal is x(n),(n=1,2,3,⋯,n), the Fourier frequency domain function obtained by Fourier transform is X(f). The cepstrum of time domain signal is obtained by taking logarithm of X(f) and then performing inverse Fourier transform. The expression of the cepstrum is shown in Equation (3).
(3)$$C_{X}\left(\tau \right)=F^{-1}\left[\ln\left(X\left(f\right)\right)\right]$$

In Equation (3), $C_{X}\left(\tau \right)$ is the cepstrum based on Fourier frequency domain function, τ is the inverse frequency, and $F^{-1}\left[\;\right]$ represents the inverse Fourier transform.

The frequency domain function obtained by fast Fourier transform can be represented in complex form. Therefore, it can be reflected by Equation (4).
(4)X(f)=a+bi

In Equation (4), a represents the real sequence of X(f), b represents the imaginary sequence of X(f), and i is the imaginary unit. According to the definition of fast Fourier transform, the amplitude and phase spectra of the signal after Fourier transform are shown in Equations (5) and (6).
(5)$$A\left(f\right)=\sqrt{a^{2}+b^{2}}$$
(6)θ(f)=arctan(b/a)

Therefore, the real cepstrum signal of the signal can be represented by Equation (7).
(7)$$C_{\left|X\right|}\left(\tau \right)=F^{-1}\left[\ln\left(\left|X\left(f\right)\right|\right)\right]=F^{-1}\left[\ln\left(A\left(f\right)\right)\right]=F^{-1}\left(\ln\left(\sqrt{a^{2}+b^{2}}\right)\right)$$

In the CPW algorithm, the mean of the logarithmic spectrum of the signal is equal to the value at the zero-reciprocal frequency in the real cepstrum. The logarithmic edited amplitude spectrum of cepstrum obtained after transformation is combined with the original phase spectrum to form a new complex spectrum as shown in Equation (8).
(8)XX(f)=aa+ibb

In Equation (8), XX(f) is the new complex spectrum, aa is the real sequence of the new complex spectrum, and bb is the imaginary part sequence of the new complex spectrum.

According to the definition and Equation (6), the expressions of aa and bb are, respectively, shown in Equations (9) and (10).
(9)$$aa=\ln\left(\bar{\ln\left(\left|X\left(f\right)\right|\right)}\right)*\cos\left(\theta \left(f\right)\right)=k\frac{a}{\sqrt{a^{2}+b^{2}}}=k\sqrt{\frac{1}{1+\tan^{2}\left(\theta \left(f\right)\right)}}$$
(10)$$bb=\ln\left(\bar{\ln\left(\left|X\left(f\right)\right|\right)}\right)*\sin\left(\theta \left(f\right)\right)=k\frac{b}{\sqrt{a^{2}+b^{2}}}=k\sqrt{\frac{1}{1+\cot^{2}\left(\theta \left(f\right)\right)}}$$

In Equations (9) and (10), because $\ln\left(\bar{\ln\left(\left|X\left(f\right)\right|\right)}\right)$ is a constant, *k* is used instead of $\ln\left(\bar{\ln\left(\left|X\left(f\right)\right|\right)}\right)$. The new complex spectrum can be further represented by Equation (11).
(11)$$XX\left(f\right)=k\left(\sqrt{\frac{1}{1+\tan^{2}\left(\theta \left(f\right)\right)}}+i\sqrt{\frac{1}{1+\cot^{2}\left(\theta \left(f\right)\right)}}\right)$$

According to the transformation formula of the trigonometric function, it can be found that the amplitude spectrum of the new complex spectrum is a constant sequence. Therefore, the complex process of calculating the logarithmic amplitude editing spectrum of cepstrum can be completely replaced by solving the constant index representing the intensity of the original time-domain signal, which can greatly simplify the CPW solution process and improve the solution efficiency.

In order to determine the intensity of the information contained in the original signal and the extent of damage to useful information in the signal, the average energy of the scalar signal is used to calculate the energy of a group of signals with reference to the Pashawar theorem [19,20]. The calculation formula is shown in Equation (12).
(12)$$S=\frac{\sum_{i=1}^{n}x{\left(t\right)}^{2}}{n}$$

Replace the *k* value in Equation (11) with the *S* value in Equation (12) and rewrite the new complex spectrum to Equation (13).
(13)$$XX\left(f\right)=S\left(\frac{a}{\sqrt{a^{2}+b^{2}}}+\frac{b}{\sqrt{a^{2}+b^{2}}}i\right)=\frac{\sum_{i=1}^{n}x{\left(t\right)}^{2}}{n}\left(\frac{a}{\sqrt{a^{2}+b^{2}}}+\frac{b}{\sqrt{a^{2}+b^{2}}}i\right)$$

### 2.3. Energy Operator Demodulation Algorithm

The energy operator [21,22] is a nonlinear difference operator. By calculating the total energy of the signal through the combination of transient value and differential form, it can highlight the transient impact information in the time domain signal, extract the impact characteristics, and realize the demodulation of the original signal [23]. It has good adaptability and time resolution. For a signal with a simple harmonic vibration response, its energy operator is defined by Equation (14) [24].
(14)$$\Psi_{k}\left[x\left(t\right),x{\left(t\right)}^{\left(k-1\right)}\right]=\dot{x}(t)x{\left(t\right)}^{\left(k-1\right)}-x\left(t\right)x{\left(t\right)}^{\left(k\right)},k=0,1,2\cdots$$

In Equation (14), $x{\left(t\right)}^{\left(k\right)}$ represents the *k*-order differential form of the time domain signal.

Generally, the energy operator of the second order (i.e., *k* = 2) is called the Teager energy operator, and its calculation formula is shown in Equation (15).
(15)$$\Psi \left[x\left(t\right)\right]=\dot{x}{\left(t\right)}^{2}-x\left(t\right)\ddot{x}\left(t\right)$$

In Equation (15), x(t) is the original time domain signal, $\dot{x}(t)$ is the first derivative of x(t), and $\ddot{x}\left(t\right)$ is the second derivative of x(t).

As the actual time domain signal collected is discrete, the differential form in Equation (15) needs to be replaced by difference equations. The discrete form of x(t) is expressed as x(n). The first derivative of x(n) can be expressed by different difference methods.
(16)$$\dot{x}(n)=x\left(n\right)-x\left(n-1\right)$$
(17)$$\dot{x}(n)=x\left(n+1\right)-x\left(n\right)$$

By substituting Equations (14)–(16) into Equation (13), we can get the following results.
(18)$$\begin{array}{l} \Psi \left[x\left(n\right)\right]\\ =\left[x\left(n\right)-x\left(n-1\right)\right]\left[x\left(n+1\right)-x\left(n\right)\right]\\ -x\left(n\right)\left[x\left(n+1\right)-2x\left(n\right)+x\left(n-1\right)\right]\\ =x{\left(n\right)}^{2}-x\left(n+1\right)x\left(n-1\right) \end{array}$$

Equation (18) is the differential expression of Teager energy operator. Through the discrete form of Teager energy operator, the general expression of discrete expression of energy operator can be deduced.
(19)$$\Psi_{k}\left[x\left(n\right)\right]=x\left(n\right)x\left(n+k-2\right)-x\left(n+k-1\right)x\left(n-1\right),k=0,1,2,\cdots$$

By calculating the energy operator of the time domain signal, the envelope demodulation of the signal can be realized and the hidden characteristic information of the signal can be extracted.

### 2.4. Calculation of Bearing Fault Characteristic Frequency

The axle box bearings of high-speed trains are mostly double row cylindrical bearings and double row tapered bearings, both of which are rolling bearings. The fault types of rolling bearing can be divided into outer ring fault, inner ring fault, rolling element fault, and cage fault. When there is damage in the inner part of the bearing, and other parts contact the damaged part, the corresponding impulse impact excitation will be generated. This kind of impact excitation is different from the normal vibration of the bearing: the duration is very short, but it has a very wide spectrum range, which can arouse the resonance of all parts of the bearing. Different parts have different impact periods due to their different motion modes, so they have different impact frequencies. This frequency is the interval frequency of local defects, which can reflect most of the faults of the bearing, so it can be called the fault frequency of the bearing. Combining with the geometric structure and speed of the bearing, the fault characteristic frequency of each component of the bearing can be calculated [25].

When the outer ring is fixed, the fault frequency calculation of the outer ring of rolling bearing is shown in Equation (20).
(20)$$f_{o}=\frac{rn}{120}\left(1-\frac{d\cos\alpha }{D}\right)$$

When the outer ring is fixed, the fault frequency calculation of inner ring of rolling bearing is shown in Equation (21).
(21)$$f_{i}=\frac{rn}{120}\left(1+\frac{d\cos\alpha }{D}\right)$$

When the outer ring is fixed, the fault frequency calculation of rolling element of rolling bearing is shown in Equation (22).
(22)$$f_{b}=\frac{rD}{120d}\left(1-{\left(\frac{d\cos\alpha }{D}\right)}^{2}\right)$$

When the outer ring is fixed, the fault frequency calculation of cage of rolling bearing is shown in Equation (23).
(23)$$f_{c}=\frac{r}{120}\left(1-\frac{d\cos\alpha }{D}\right)$$

In Equations (20)–(23), r is the rotational speed of the rolling bearing, r/min; n is the number of rolling elements of the bearing; d is the diameter of the rolling element, mm; D is the diameter of the pitch circle of the rolling bearing, mm; and α is the contact angle of the rolling element.

By substituting the geometric parameters of rolling bearing into the fault frequency calculation formula, the fault frequency of each part of the bearing can be calculated.

## 3. Fault Feature Extraction Process

This section may be divided by subheadings. It should provide a concise and precise description of the experimental results, their interpretation, as well as the experimental conclusions that can be drawn.

According to the vibration time domain data of rolling bearing, this paper proposes the following fault diagnosis method flow to determine the location of bearing fault. The specific steps are as follows.
(1)Filter the original time domain signal by AR model filter.(2)Remove the average value of filtered signal to eliminate the influence of DC signal.(3)Calculate the average energy values of the signal.(4)Whiten the time domain signal and combine with the average energy value to form a new complex spectrum.(5)Solve the energy operator of the new complex spectrum to demodulate the envelope and enhance the feature.(6)Use the energy operator to analyze the frequency spectrum, extract the characteristic frequency, and compare with the bearing fault characteristic frequency to determine the fault location of the bearing.

The flow chart of fault feature extraction of rolling bearing is shown in Figure 4.

## 4. Data Analysis and Verification

### 4.1. Simulation Data Analysis

In order to verify the effectiveness of the fault diagnosis process of axle box bearing proposed in this paper, the rolling bearing fault model is used for data modeling. The expression of simulation signal is shown in Equation (24):(24)x(t)=s(t)+n(t)

In Equation (24), s(t) represents periodic shock signal and n(t) represents complex strong noise generated by bearing operating environment. The expression of s(t) is as follows [26,27,28].
(25)$$\{\begin{array}{c} s\left(t\right)=\sum_{i=1}^{N}A_{i}h_{i}\\ A_{i}=1+A_{0}\sin\left(2\pi f_{r}t\right)\\ h_{i}=e^{-C\left(t-iT-\tau_{i}\right)}\sin\left[2\pi f_{n}\left(t-iT-\tau_{i}\right)\right] \end{array}$$

In Equation (25), N is the number of fault shocks; $A_{0}$ is the initial value of amplitude; $f_{r}$ is the bearing rotation frequency; C is the impact attenuation coefficient; T the time interval of adjacent shocks (reciprocal of fault characteristic frequency); $\tau_{i}$ the time error of mutual movement of internal parts of bearing, usually 1–2% of T; and $f_{n}$ is the resonance frequency of bearing system.

According to the parameter data in Table 1, the vibration data of inner ring fault of rolling bearing is established.

Suppose the sampling frequency of the system is 12,000 Hz, the time length is 0.5 s, and the fault frequency is 291. The simulation signal is shown in Figure 5.

Figure 5a shows the waveform of noiseless simulation signal, which shows obvious periodic shock waveform; Figure 5b shows the waveform of simulation signal with noise. It can be seen that no obvious periodic shock component can be seen after adding noise. The frequency domain diagram is shown in Figure 6.

From Figure 6, it can be seen that there is a higher frequency amplitude near the resonance frequency, but there is no obvious frequency amplitude at the fault frequency and its multiple frequency in the whole spectrum range. The time domain signal is processed according to the fault diagnosis flow of this paper, and the processing results are shown in Figure 7.

It can be seen from Figure 7f that after the signal passes through the fault diagnosis process in this paper, the amplitude of frequency spectral lines at bearing rotation frequency $f_{r}$, fault frequency $f_{i}$, and its second harmonic generation $2f_{i}$, and third harmonic generation $3f_{i}$ are very prominent. Therefore, it can be judged that the frequency domain fault characteristics of the noise-containing simulation signal have appeared, and the fault diagnosis identification has been realized.

### 4.2. Analysis and Verification of Bearing Experimental Data of Western Reserve University

This paper selects three sets of bearing fault data from the bearing data center of Western Reserve University to verify the diagnosis process. The experimental table is shown in Figure 8 [29,30].

The selected bearing data are the fault data of a drive-end bearing at 12 K sampling frequency, and the bearing signal is from a 6205-2RS JEM SKF deep groove ball bearing. The parameters of the bearing are shown in Table 2.

The selected three groups of fault data parameters are shown in Table 3.

The processing result of fault data a is shown in Figure 9.

In Figure 9, the selected signal is the fault data of bearing inner ring. The fault characteristics of the data cannot be directly found from the signal time domain diagram and frequency domain diagram. After pre-whitening treatment and energy operator demodulation, the spectral line amplitude at 157.5 Hz in Figure 9c is very prominent, and the spectral lines at 315 Hz, 471.9 Hz, and 944.3 Hz are also prominent. According to the characteristics of bearing inner ring in Equation (21), the signal cannot be directly identified. According to the eigenfrequency formula, the fault frequency of bearing inner ring is 157.9 Hz, which is very close to the main frequency 157.5 Hz in frequency domain diagram, and the error is within reasonable range. Therefore, it can be considered that the fault diagnosis method can identify the inner ring fault of roller bearing.

The processing result of fault data B is shown in Figure 10.

The data in Figure 10 are the fault data of bearing outer ring. According to the characteristic frequency formula of the bearing outer ring in Equation (20), the fault frequency of bearing outer ring is 107.4 Hz, which is close to the prominent spectral line 107.5 Hz in Figure 10c, and there are obvious peak spectral lines at the second, third, fourth, fifth, and sixth harmonic generation, so the bearing can be effectively identified the failure of the outer ring.

The processing result of fault data C is shown in Figure 11.

The data in Figure 11 are the fault data of bearing rolling element, which are calculated according to the characteristic frequency formula of bearing rolling element in Equation (22). Under this working condition, the fault frequency of bearing rolling element is 139.2 Hz, and the characteristic frequency in Figure 11c is 137.5 Hz. There is also a more obvious peak value of spectral line at the second harmonic generation, which can identify the fault of bearing rolling element.

### 4.3. Analysis and Verification of Test Data of Axle Box Bearing of High-Speed Train

In order to further verify the effectiveness of the fault diagnosis method proposed in this paper, the bearing vibration acceleration signal collected by the axle box bearing testbed of high-speed train is used for analysis and verification. The testbed is shown in Figure 12.

The testbed is composed of a support bearing, tested bearing, and vibration exciter, and can exert different static load on the tested bearing. Different operating conditions are simulated by applying different rotating speed, static load, and excitation frequency, and the bearing monitoring data (including vibration data, temperature data, and sound data) of different fault types under various working conditions are collected. The test bearing used in the testbed is the axle box bearing of a domestic electric multiple units (EMU). The calculation results of the size parameters and fault characteristic order ratio (characteristic order ratio is the ratio of fault characteristic frequency to rotation frequency) are shown in Table 4.

The vibration acceleration signals of the axle box bearing under two different working conditions are analyzed. The photo of faulty axle box bearing is shown in Figure 13.

Figure 13 shows two kinds of fault bearing with different working conditions. The vibration data are collected by National Instruments (NI) vibration three-phase sensors and corresponding data acquisition modules. The parameters of two kinds of working conditions are shown in Table 5.

The vibration data of the two working conditions are analyzed, the time domain diagrams corresponding to the two conditions are shown in Figure 14.

In order to reduce the amount of data and increase the data samples at the same time, six sections of data are selected from the fault time domain signal of the outer ring of Condition No.1. Each group of data is randomly selected for 5 s, and the time domain diagram of segmented data is shown in Figure 14.

For each group of data, the axle box bearing fault diagnosis method is used for feature extraction, and the result of feature extraction is shown in Figure 15.

The data in Figure 16 is the vibration acceleration data of axle box bearing outer ring fault. According to the bearing dimension parameters and Equation (20), under this condition, the fault frequency of bearing outer ring is 97.35 Hz. In Figure 15, the characteristic frequency of each group of data is 98 Hz, and there are obvious spectral peaks at the second and third harmonic generation, which can clearly see the three orders of bearing outer ring fault characteristic frequency. Therefore, it can be considered that this method can effectively identify the outer ring fault of axle box bearing.

Six sections of data are selected from the fault time domain signal of the rolling element of Condition No.2. Each group of data is randomly selected for 5 s, and the time domain diagram of segmented data is shown in Figure 17.

For each group of data, the axle box bearing fault diagnosis method is used for feature extraction, and the result of feature extraction is shown in Figure 18.

The data in Figure 18 are the vibration acceleration data of the axle box bearing rolling element fault. According to the bearing dimension parameters and Equation (22), under this condition, the fault frequency of bearing outer ring is 62.29 Hz. In Figure 17, the characteristic frequency of each group of data is 61.4 Hz, and there are obvious spectral peaks at the second, third, and fourth harmonic generation, which can clearly see the four orders of bearing rolling element fault characteristic frequency. Therefore, it can be considered that this method can effectively identify the rolling element fault of axle box bearing.

## 5. Discussion and Conclusions

In this paper, the axle box bearing of rail train bogie is taken as the research object, and the vibration signal is used as the data source for bearing fault diagnosis. By filtering and whitening the vibration data, the energy operator of the processed signal is extracted, and then the frequency spectrum is analyzed. Finally, the fault characteristic frequency can be obtained. Through the analysis of the bearing vibration simulation data and experimental data, the effectiveness of the fault diagnosis process is verified. The conclusions are as follows.
(1).The original vibration signal is processed by AR filtering, spectrum whitening, and energy operator, which can effectively remove the redundant and complex frequency components in the data, realize data denoising, improve the signal-to-noise ratio (SNR) of data, highlight the characteristic frequency of data, and have high calculation efficiency.(2).The diagnosis process has good adaptive characteristics, which can avoid the setting and selection of parameters in some fault diagnosis methods, and is suitable for monitoring and diagnosing a large number of vibration data of rail transit vehicle system.(3).However, through the analysis of a large amount of experimental data, it is found that the fault diagnosis process proposed in this paper has a high diagnosis accuracy rate for the inner and outer rings of rolling bearing, but the diagnosis accuracy rate for rolling element fault is relatively low. More than 100 sets of vibration data under different operating conditions were collected for bearings with different faults; the recognition rate of outer ring fault data is 90.3%, the recognition rate of outer ring fault data is 86.1%, and the recognition rate of rolling element fault data is 72.2%. There is always a characteristic frequency error of 1–2 Hz, which may be caused by the speed error. At the same time, when the speed is relatively low, the fault frequency will be covered by the transfer frequency. In order to improve the accuracy of rolling element fault diagnosis, it is necessary to optimize the fault diagnosis method.

## Figures and Tables

**Figure 1 sensors-20-07155-f001:**
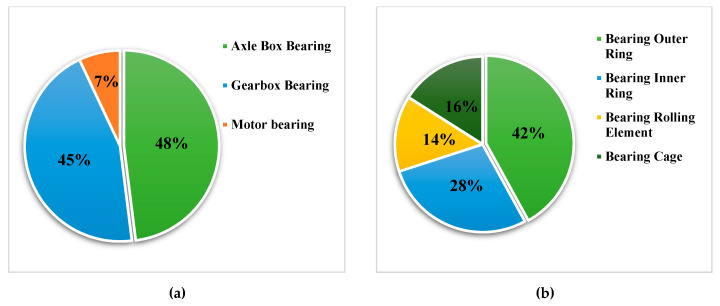
Statistics of bogie bearing failure. (**a**) The type of fault bearing. (**b**) The faulty part of the bearing.

**Figure 2 sensors-20-07155-f002:**
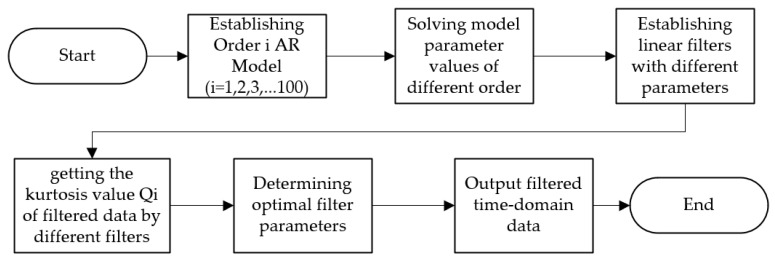
Autoregressive (AR) model filter filtering process.

**Figure 3 sensors-20-07155-f003:**
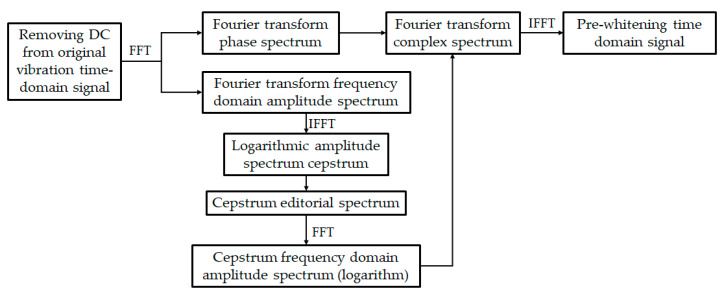
Flow chart of the Cepstrum Editing Pre-Whitening (CPW) algorithm.

**Figure 4 sensors-20-07155-f004:**
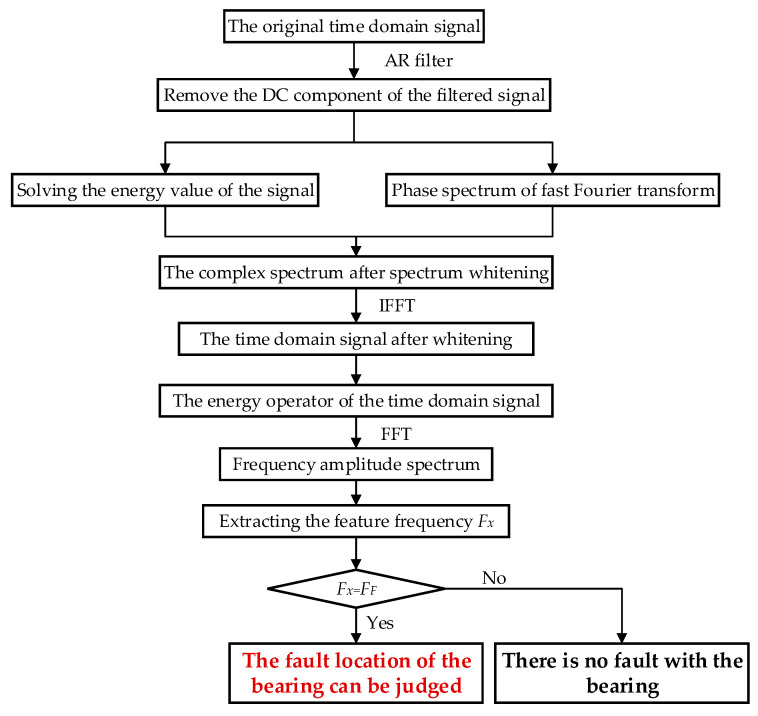
Flow chart of fault diagnosis.

**Figure 5 sensors-20-07155-f005:**
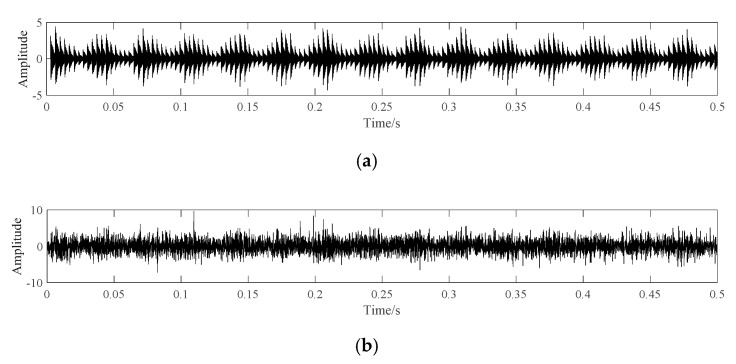
The Waveform of simulation signal. (**a**) The simulation signal without noise. (**b**) The simulation signal with noise.

**Figure 6 sensors-20-07155-f006:**
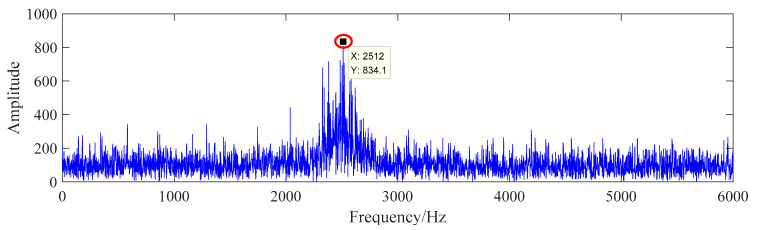
The frequency domain of the signal with noise.

**Figure 7 sensors-20-07155-f007:**
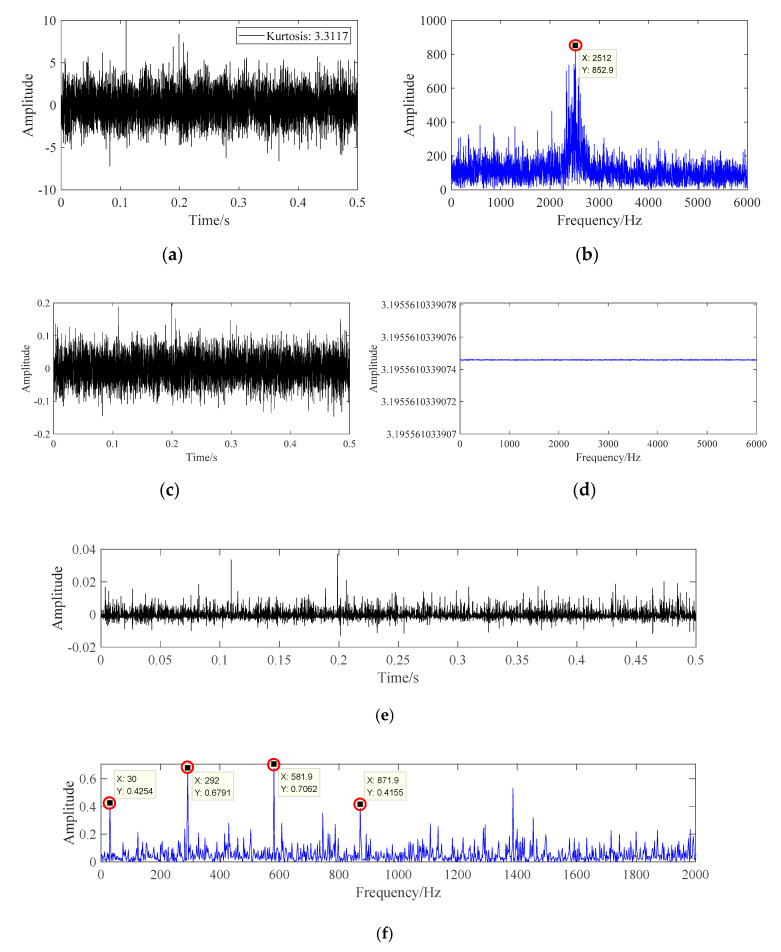
AR filtering signal diagram. (**a**) Time domain diagram of AR filtering signal. (**b**) Frequency domain diagram of AR filtering signal. (**c**) Time domain diagram of whitening signal. (**d**) Frequency domain diagram of whitening signal (**e**) The energy operator spectrum of whitening signal. (**f**) Frequency spectrum of the energy operator.

**Figure 8 sensors-20-07155-f008:**
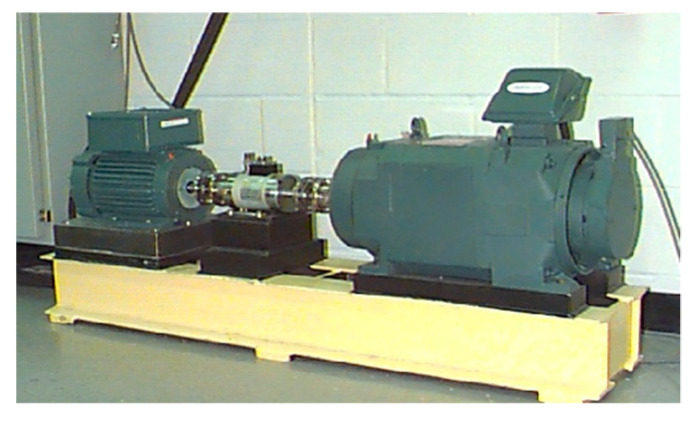
The experimental station of rolling bearing.

**Figure 9 sensors-20-07155-f009:**
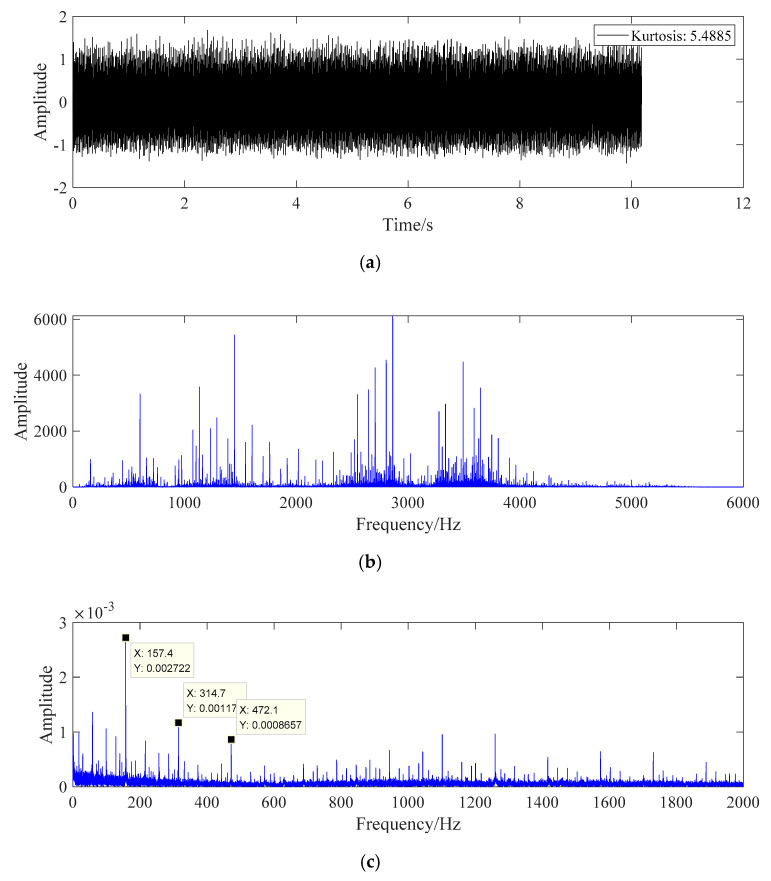
The fault diagnosis diagram of fault data A. (**a**) Time domain diagram of fault data A. (**b**) Frequency domain diagram of fault data A. (**c**) Fault data A whitening energy operator frequency domain graph.

**Figure 10 sensors-20-07155-f010:**
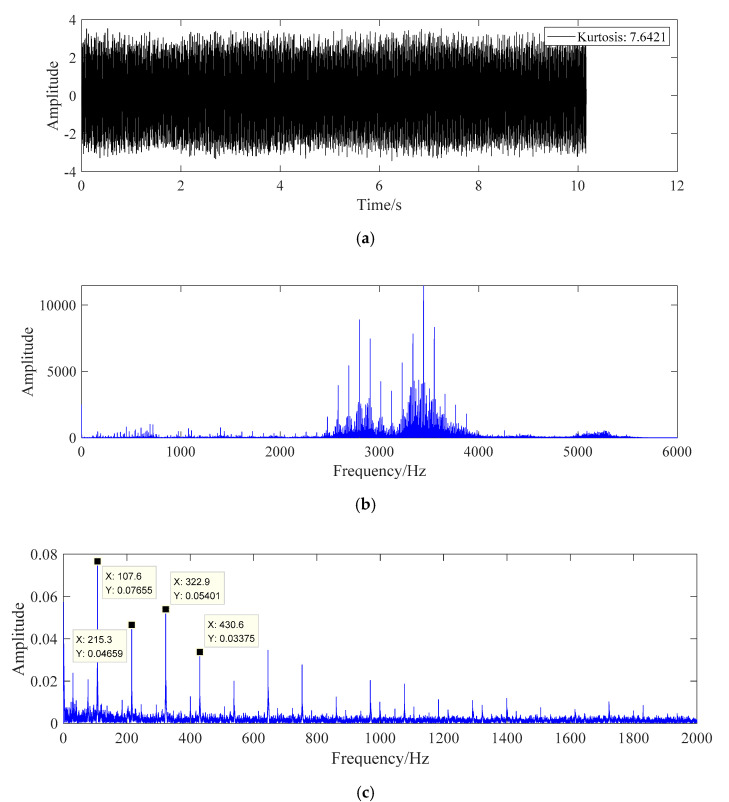
The fault diagnosis diagram of fault data B. (**a**) Time domain diagram of fault data B. (**b**) Frequency domain diagram of fault data B. (**c**) Fault data B whitening energy operator frequency domain graph.

**Figure 11 sensors-20-07155-f011:**
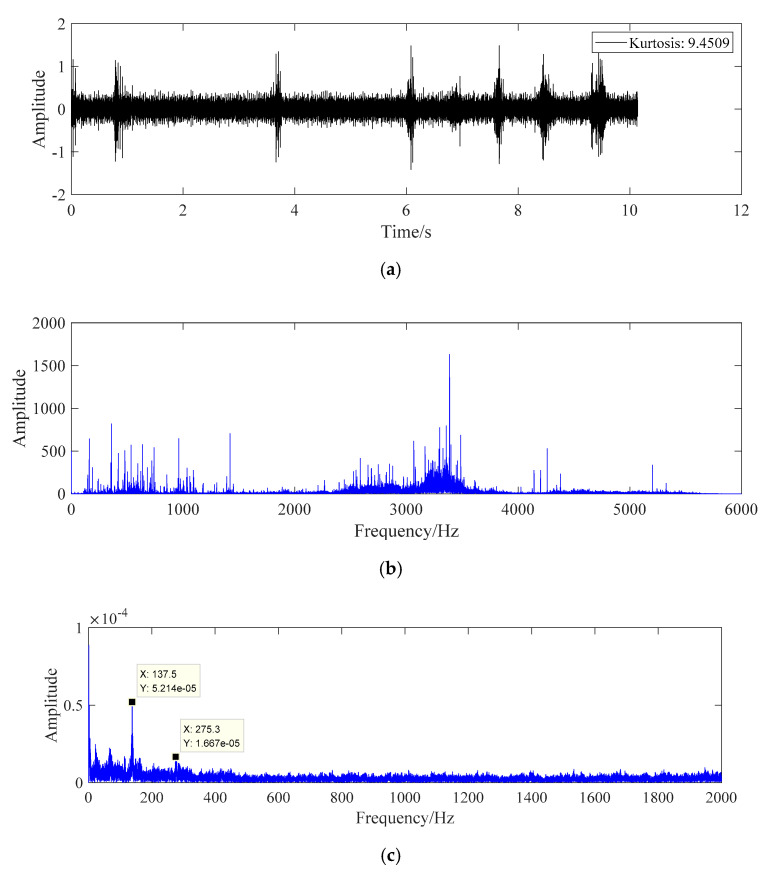
The fault diagnosis diagram of fault data C. (**a**) Time domain diagram of fault data C. (**b**) Frequency domain diagram of fault data C. (**c**) Fault data C whitening energy operator frequency domain graph.

**Figure 12 sensors-20-07155-f012:**
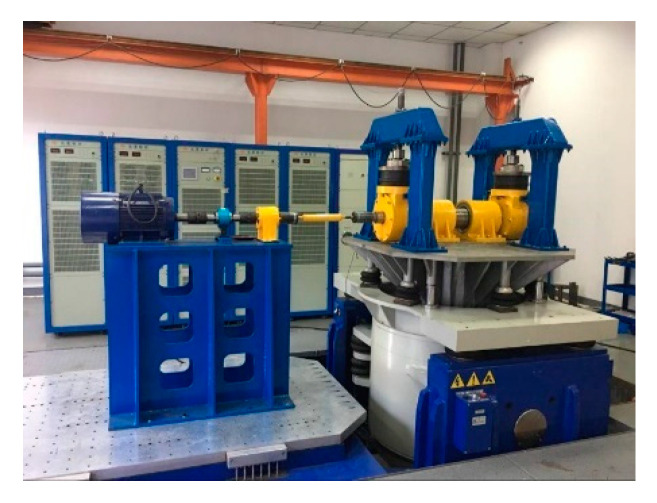
The experimental station of high-speed train axle box bearing.

**Figure 13 sensors-20-07155-f013:**
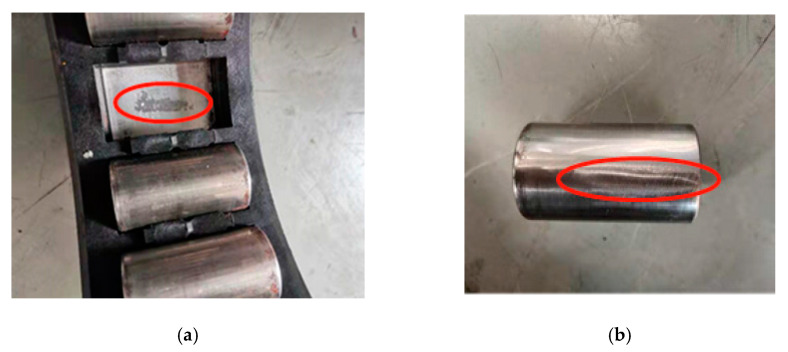
The photo of faulty axle box bearing. (**a**) Bearing outer ring failure. (**b**) Bearing rolling element failure.

**Figure 14 sensors-20-07155-f014:**
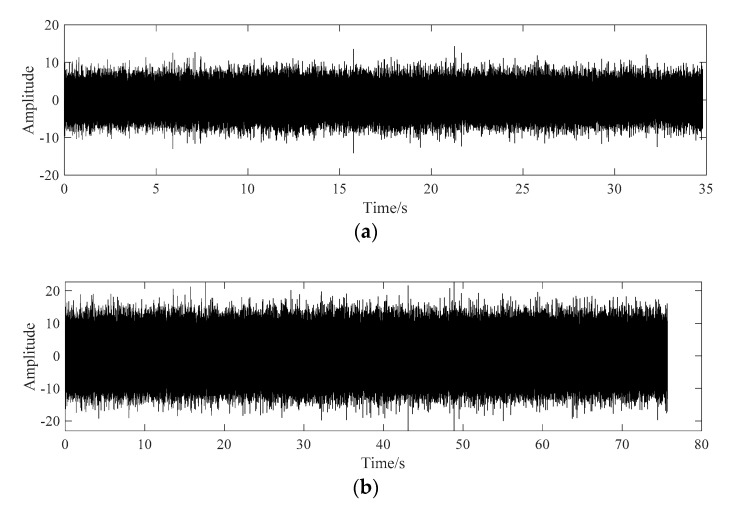
The domain diagram of fault bearings. (**a**) Time domain diagram of outer ring fault. (**b**) Time domain diagram of rolling element fault.

**Figure 15 sensors-20-07155-f015:**
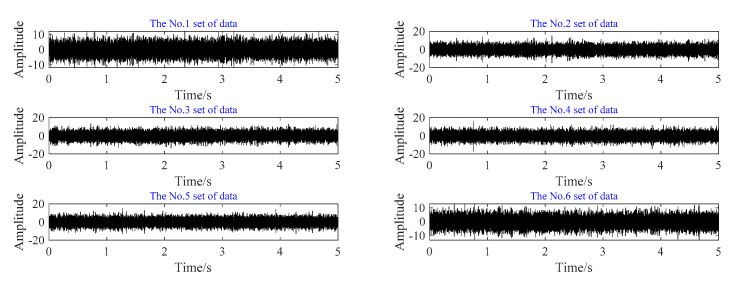
Sectional time domain diagram of outer ring fault.

**Figure 16 sensors-20-07155-f016:**
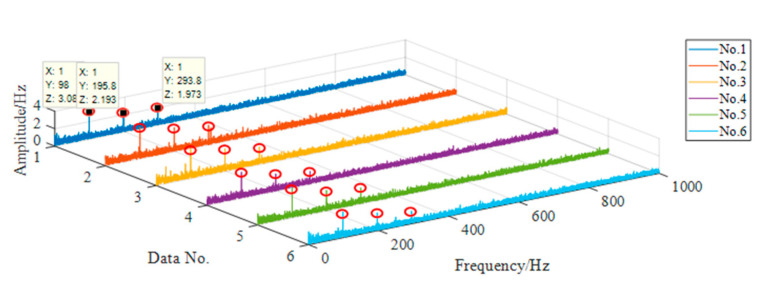
Characteristic frequency distribution of outer ring fault.

**Figure 17 sensors-20-07155-f017:**
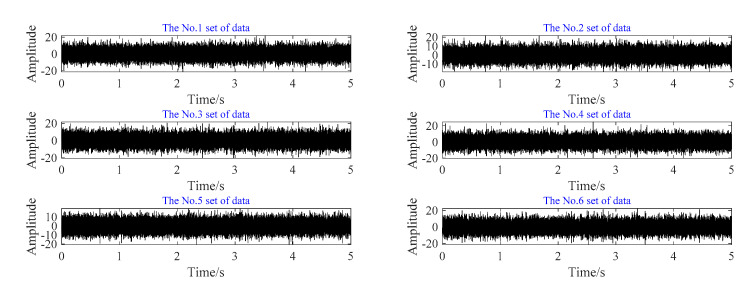
Sectional time domain diagram of rolling element fault.

**Figure 18 sensors-20-07155-f018:**
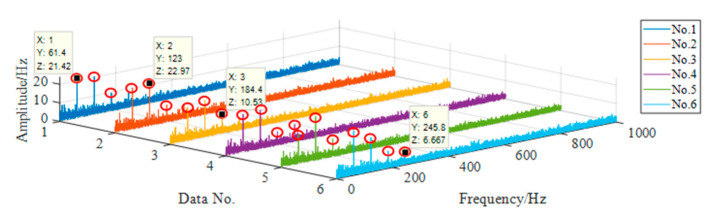
Characteristic frequency distribution of rolling element fault.

**Table 1 sensors-20-07155-t001:** Simulation parameters of rolling bearing fault model.

Simulation Variables	Parameter Value
Number of fault shocks N	145
Initial value of amplitude $A_{0}$	0.5
Bearing rotation frequency $f_{r}$	30 Hz
Shock attenuation coefficient C	500
Time interval between adjacent shocks T	(1/291) s
Resonance frequency of bearing system $f_{n}$	2500 Hz

**Table 2 sensors-20-07155-t002:** Dimension parameters table of the rolling bearing.

Rolling Bearing Parameters	Parameter Value
Bearing inner ring diameter	25 mm
Bearing outer ring diameter	52 mm
Bearing thickness	15 mm
Bearing rolling element diameter	7.94 mm
Bearing rolling element diameter	39.04 mm
Number of scrolls	9

**Table 3 sensors-20-07155-t003:** Parameters table of fault data.

Serial Number of Fault Data	A	B	C
Fault location	Bearing inner ring	Bearing outer ring	Bearing rolling element
Bearing speed (r/min)	1750	1797	1772
Fault diameter (mm)	0.1778	0.1778	0.5334

**Table 4 sensors-20-07155-t004:** Dimension and fault characteristic parameters table of the axle box bearing.

Diameter of axle box bearing pitch circle (mm)	185
Rolling element diameter (mm)	26.68
Number of single column scrolls	17
Contact angle (°)	12.083
Characteristic order ratio of outer ring fault	7.3013
Fault characteristic order ratio of rolling element	3.3981
Cage fault characteristic order ratio	0.42949

**Table 5 sensors-20-07155-t005:** Working condition table of axle box bearing testbed.

Condition No.	Condition No.1	Condition No.2
Fault location	Bearing outer ring	Bearing rolling element
Bearing speed (r/min)	800	1100
Sampling frequency (Hz)	25,600	25,600
Static load force (kg)	1000	500
Basic frequency of vibration exciter	0	10
Fault location	Bearing outer ring	Bearing rolling element

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
