# Peer review of "A Fault Diagnosis Method of Bogie Axle Box Bearing Based on Spectrum Whitening Demodulation"

_sensors, 2020, doi:10.3390/s20247155_

Round 1
Reviewer 1 Report
This article presents a fault diagnosis method of bogie axle box bearing based on spectrum whitening demodulation. It is very interesting and well organized. However, it has some problems that need to be solved further.
1. In the abstract, the authors describe the research background and significance, which is very good. Moreover, the important details of the proposed method are given. However, the author does not explain why such a method is used and what its innovation is.
2. Many formulas have problems, such as Line 110, 111, 145, 183, 227, 230, 233, 236.
3. In the introduction, the introduction of fault diagnosis background is a little insufficient, some of newly developed methods are not mentioned.
Wang, X.; Lu, Z.; Wei, J.; Zhang, Y. Fault Diagnosis for Rail Vehicle Axle-Box Bearings Based on Energy Feature Reconstruction and Composite Multiscale Permutation Entropy. Entropy 2019, 21, 865.
Song, E.; Ke, Y.; Yao, C.; Dong, Q.; Yang, L. Fault Diagnosis Method for High-Pressure Common Rail Injector Based on IFOA-VMD and Hierarchical Dispersion Entropy. Entropy 2019, 21, 923.
4. In conclusions, the authors should provide main numerical results to prove the proposed method.
5. All references are not in standard format. The authors should check and revise them carefully.
In order to qualify for publication in Sensors, the paper must be improved according to the comments to the authors.
Reviewer 2 Report
Dear authors,
congratulations for your work. It seems that this method could be really useful in fault bearing detection.
Here are my comments:
Abstract:
- Why have you include the numbers into brakets?
Introduction:
- Line 39: There is an odd “f” at the end of the line.
- Figure 1a: The text “7%” can’t be properly seen.
- Figure 1b: I think that its title is not correct. It is not “the location of the faulty bearing”. I think that it is “the location of the fault on the bearing” or “the faulty part of the bearing”.
- Line 48: You should include the meaning of all acronims, such as TADS.
- Line 51: You should include the meaning of all acronims, such as LMLSTM.
- Line 62: You should include the meaning of all acronims, such as EEMD.
- Line 67: You should include the meaning of all acronims, such as MCKD.
Section 2:
- Equation 2: The subindex of the elemento on the second column of the third row is wrong. I think that it should be a “1”, instead of a “0”.
- Line 125: You should use a capital “W”, in “Pre-withening”.
- Line 139: A discrete time-domain signal is expressed directly as x[n].
- Equation 3: It seems that you are calculating the logarithm of the whole X(f) Fourier transform, not the logarithm of the amplitude of the X(f) Fourier transform.
- Equations 9 and 10: What is the meaning of “k”? You should explain it after the equations.
- Line 172: “Is a series constant”? I don’t understand what do you mean with this part of the sentence.
- Lines 197-198: The differential form needs to be replace by differential form? Is this sentence ok?
- Line 237: In Equations.
Section 3:
- Line 251: “values”. There is an extra gap between “value” and “s”.
Section 4:
- Line 278: You haven’t written the units of the sample frequency. And you haven’t written the value of the time length.
- Table 3: Last row don’t show bearing speed values, do it?
- Line 384: Include the meaning of EMU.
Yours sincerely
Round 2
Reviewer 1 Report
Thank the authors for their efforts. The authors have adequately addressed all my concerns in the review, and did a good job to revise and improve the paper. The paper now is suitable for publication in Sensors in its current form.